# The DEAD-box RNA helicase Dhx15 controls glycolysis and arbovirus replication in *Aedes aegypti* mosquito cells

**Samara Rosendo Machado[1], Jieqiong Qu[1], Werner J. H. Koopman[2], Pascal Miesen[1]\***

**1** Department of Medical Microbiology, Radboud Institute for Molecular Life Sciences, Radboud University Medical Center, Nijmegen, The Netherlands, **2** Department of Pediatrics, Amalia Children's Hospital, Radboud Institute for Molecular Life Sciences, Radboud Center for Mitochondrial Medicine, Radboud University Medical Center, Nijmegen, The Netherlands

\* pascal.miesen@radboudumc.nl

## Abstract

*Aedes aegypti* mosquitoes are responsible for the transmission of arthropod-borne (arbo) viruses including dengue and chikungunya virus (CHIKV) but in contrast to human hosts, arbovirus-infected mosquitoes are able to efficiently control virus replication to sub-pathological levels. Yet, our knowledge of the molecular interactions of arboviruses with their mosquito hosts is incomplete. Here, we aimed to identify and characterize novel host genes that control arbovirus replication in *Aedes* mosquitoes. RNA binding proteins (RBPs) are well-known to regulate immune signaling pathways in all kingdoms of life. We therefore performed a knockdown screen targeting 461 genes encoding predicted RBPs in *Aedes aegypti* Aag2 cells and identified 15 genes with antiviral activity against Sindbis virus. Amongst these, the three DEAD-box RNA helicases AAEL004419/Dhx15, AAEL008728, and AAEL004859 also acted as antiviral factors in dengue and CHIKV infections. Here, we explored the mechanism of Dhx15 in regulating an antiviral transcriptional response in mosquitoes by silencing *Dhx15* in Aag2 cells followed by deep-sequencing of poly-A enriched RNAs. *Dhx15* knockdown in uninfected and CHIKV-infected cells resulted in differential expression of 856 and 372 genes, respectively. Interestingly, amongst the consistently downregulated genes, *glycolytic process* was the most enriched gene ontology (GO) term as the expression of all core enzymes of the glycolytic pathway was reduced, suggesting that Dhx15 regulates glycolytic function. A decrease in lactate production indicated that *Dhx15* silencing indeed functionally impaired glycolysis. Modified rates of glycolytic metabolism have been implicated in controlling the replication of several classes of viruses and strikingly, infection of Aag2 cells with CHIKV by itself also resulted in the decrease of several glycolytic genes. Our data suggests that Dhx15 regulates replication of CHIKV, and possibly other arboviruses, by controlling glycolysis in mosquito cells.

**Data Availability Statement:** All deep sequencing files are available from NCBI sequence read archive (SRA) under the Bioproject accession number PRJNA885496. All numeric data and images are

published as source data along with the manuscript (S4 Table) and is deposited in DANS-Easy, an open access data depository operated by the Royal Dutch Academy of Sciences (KNAW) under the DOI https://doi.org/10.17026/dans-ze4-zf72.

**Funding:** This work was financially supported by a Veni grant (ID: VI.Veni.202.035) from the Dutch Research Council (Nederlandse Organisatie voor Wetenschappelijk Onderzoek; https://www.nwo.nl/) to PM. The funders had no role in study design, data collection and analysis, decision to publish, or preparation of the manuscript.

**Competing interests:** The authors have declared that no competing interests exist.

## Author summary

*Aedes aegypti* mosquitoes transmit pathogenic human arthropod-borne (arbo)viruses including Chikungunya (CHIKV) and dengue virus, however, the molecular mechanism that control virus infection in mosquitoes are poorly understood. Here, we aimed to discover novel host genes involved in restricting virus growth in *Aedes* mosquitoes. We therefore individually inactivated 461 genes in mosquito cells and assessed the replication of Sindbis virus, an arbovirus from the same virus family as CHIKV. We discovered 15 antiviral genes that, when silenced, resulted in increased virus replication. Amongst these was Dhx15, a member of the DEAD-box RNA helicases, which are well-known for their diverse roles in regulating immune responses. To understand the mechanism that underlie the antiviral activity of Dhx15, we measured gene expression in mosquito cells depleted of Dhx15. We observed that genes that control sugar metabolism via the glycolysis pathway were downregulated, which indeed resulted in a functional impairment of glycolysis. Interestingly, CHIKV infection of mosquito cells resulted in a similar downregulation of glycolytic genes, underscoring the importance of sugar metabolism in regulating virus infections. The role of metabolic pathways in shaping immune responses is increasingly recognized and our study provides a novel perspective on how such metabolic responses are regulated in vector mosquitoes.

## Introduction

The yellow fever mosquito *Aedes aegypti* is the principal vector of numerous medically important arthropod-borne viruses (arboviruses) including Chikungunya virus (CHIKV; genus *Alphavirus*, family *Togaviridae*) and dengue virus (DENV; genus *Flavivirus*, family *Flaviviridae*) [1–3]. CHIKV and DENV infections cause similar, flu-like symptoms including headache, fever, and muscle pain. Yet, more serious CHIKV infections manifest with severe joint pain and arthritis that sometimes persist for weeks up to years [4,5], whereas serious DENV infections may result in hemorrhagic fever [1]. *Ae. aegypti* mosquitoes were originally restricted to (sub)tropical countries. However, elevated global temperatures, increased urbanization, and more extensive international travel and trade have favored mosquito invasion into more temperate climate zones [1]. The expansion of the *Ae. aegypti* habitat has consequently lead to the global spread of arboviruses alike [6].

The ability of mosquitoes to acquire, replicate, and transmit arboviruses, collectively referred as vector competence, is a key determinant for efficient arbovirus transmission [7]. Upon acquisition in an infected bloodmeal, viruses initially infect midgut epithelial cells and subsequently disseminate to secondary tissues. Once a systemic infection is established and high viral titers are reached in the mosquito saliva, arbovirus transmission can take place [7,8]. Interestingly, virus accumulation in mosquitoes generally remains sub-pathological [9], suggesting that mosquitoes are able to efficiently reduce virus replication (resistance) and/or prevent virus-induced tissue damage (tolerance) [10]. However, to date, a comprehensive picture of the molecular processes that control arbovirus replication in the mosquito host is still lacking [11,12].

The fruit fly *Drosophila melanogaster*, a well-established genetic model organism, has been instrumental in dissecting the genetic basis of antiviral immunity in insects [13–15]. In *Drosophila*, the RNA interference (RNAi) pathway has been established as an important antiviral immune pathway that restricts both RNA and DNA viruses [16,17]. Studies in mosquitoes have confirmed the broad antiviral activity of this pathway across dipteran insects [11].

Moreover, work in *Drosophila* has shown that transcriptional responses through inducible immune signaling pathways, in particular the JAK-STAT (Janus kinase-signal transducers and activators of transcription) pathway and the two NFκB (Nuclear factor κB)-related Toll and IMD (immune deficiency) pathways contribute to antiviral immunity [18–20]. Whereas RNAi destroys viral RNA directly, the transcriptional regulation of immune responses has been proposed to upregulate anti-microbial peptides [21] or modulate metabolic responses [22], but in general, the role of transcriptional responses in antiviral immunity in mosquitoes is still understudied [11].

Here, we set out to identify new genetic determinants that control mosquito immune responses focusing on RNA binding proteins (RBPs), which regulate signaling pathways in response to infection in all kingdoms of life [23–27]. In particular, DEAD-box RNA helicases, a subgroup of RBPs [28], comprise well-known examples of enzymes that recognize viral RNA and modulate antiviral signaling [23–25]. These include the cytoplasmic viral RNA sensors RIG-I (retinoic-acid-inducible gene I) and MDA5 (melanoma-differentiation-associated gene 5), which are key activators of interferon signaling in vertebrates [24], and the antiviral RNAi effector Dicer 2 [29]. In addition, many more DEAD-box RNA helicases act as co-receptors and signaling intermediates in diverse immune pathways [26,27]. Due to the important and versatile role of RBPs, we deemed it likely that members of this family control arbovirus replication in vector mosquitoes.

To identify RBPs that interfere with arboviruses replication in mosquitoes, we performed a knockdown screen in *Ae. aegypti* Aag2 cells and assessed virus replication of a Sindbis reporter virus (SINV; genus *Alphavirus*, family *Togaviridae*). This approach uncovered fifteen antiviral genes that upon knockdown enhanced virus replication; amongst these, three DEAD-box RNA helicases, AAEL004419, AAEL008728, and AAEL004859 had broad antiviral activity against SINV, CHIKV, and DENV. We further characterized the mechanism underlying the antiviral activity of AAEL004419, the mosquito orthologue of Dhx15. Knockdown of this helicase decreased the expression of genes involved in glycolysis and consequentially reduced lactate production in mosquito cells. Glycolysis is a key process in energy metabolism by converting glucose into pyruvate, which is taken up by the mitochondria, oxidized to acetyl-CoA, and further metabolized in the tricarboxylic acid (TCA) cycle. Under anaerobic conditions, pyruvate can be converted into lactate, which is released from the cell [30]. Besides energy production, glycolysis provides the precursors for essential biomolecules including nucleotides, amino acids, and glycolipids/proteins [30,31]. The activity of glycolysis has direct effect on antiviral responses and has been reported to change upon infection with distinct viruses [32,33]. In line with this notion, we show that CHIKV infection of Aag2 cells reduced the expression of several glycolysis related genes, similar to knockdown of *AAEL004419/Dhx15*. This crosstalk at the level of glycolytic gene expression suggests that AAEL004419/Dhx15 controls CHIKV infection by regulating the glycolysis pathway in mosquito cells.

## Materials and methods

### RNA binding protein selection

Genes encoding RBPs were selected based on gene annotations from VectorBase release 2017–8 that used the *Ae. aegypti* L3 genome as reference. Using the Biomart-plugin, genes associated with the gene ontology (GO) term "RNA binding" (GO:0003723) were selected from the *Ae. aegypti* gene dataset. This analysis was repeated for four additional dipteran species with annotated genomes: *Ae. albopictus*, *Culex quinquefasciatus*, *Anopheles gambiae*, and *Drosophila melanogaster*. For the predicted RNA binding proteins from these species, *Ae. aegypti* orthologues were identified using the Biomart functionality within VectorBase and all lists of genes were

combined into a non-redundant set of genes encoding putative RNA binding proteins. We manually excluded genes that were unambiguously annotated as part of the core transcriptional, translation, and splicing machineries. The remaining genes were included in the RNAi screen and selected for double-stranded RNA (dsRNA) production and knockdown in Aag2 cells (S1 Table).

Of note, retrospective manual inspection of the candidate genes included in the screen identified a few genes that, based on the protein feature annotation in VectorBase, did not contain RBP domains. This may be due to the revisited genome annotation or the orthologue-conversion step which may define an *Ae. aegypti* orthologue that lacks RBP domains. Also, due to several updates of the *Ae. aegypti* reference genome annotation, some genes initially selected have been discontinued from the database or the annotation has been changed. Throughout the manuscript, the current gene identifiers of the L5 version of the *Ae. aegypti* genome are used. *NB*: The Biomart-function within VectorBase has been discontinued and replaced with a different search interface.

## Cells and viruses

*Ae. aegypti* Aag2 cells and the Aag2 C3PC12 clone [34] derived from these cells (cleared of the persistently infecting viruses Cell fusing agent virus, Phasi Charoen-like virus and Culex Y virus) were maintained at 28°C in Leibovitz's L-15 medium (Invitrogen: catalogue number: 21083027) supplemented with 10% foetal bovine serum (Gibco), 50 U/mL penicillin, 50 μg/mL streptomycin (Gibco), 2% tryptose phosphate broth (Sigma), and 1% non-essential amino acids (Gibco). For lactate assays, Aag2 C3PC12 cells were cultured in Schneider's *Drosophila* medium (Invitrogen, catalogue number 21720024) containing 11.11 mM D-glucose and 12.32 mM L-glutamine. This medium was supplemented with 10% foetal bovine serum (Gibco), 50 U/mL penicillin, and 50 μg/mL streptomycin (Gibco). Hela cells, BHK-15, and BHK-21 cells were maintained at 37°C, 5% $CO_2$ in Dulbecco's modified Eagle medium (DMEM) containing 25 mM D-glucose, 4 mM L-glutamine, and 1 mM sodium pyruvate (Life Technologies, catalogue number 11995065). This medium was supplemented with 10% foetal bovine serum (Gibco), 50 U/mL penicillin, and 50 μg/mL streptomycin (Gibco).

The infectious SINV-nLuc clone, expressing a nano-luciferase (nLuc) reporter as fusion protein with non-structural protein 3 (nsP3), was grown in BHK-21 cells as previously described [35]. The CHIKV expression plasmid encoding the Leiden synthetic (LS3) wildtype strain [36] was kindly provided by Dr. M.J. van Hemert (Leiden University Medical Center) and viral RNA was obtained by *in vitro* transcription on linearized plasmids using T7 mMessage mMachine (Invitrogen). RNA was then transfected into BHK-21 to grow infectious virus. Stocks of DENV serotype 2 (New Guinea C [NGC] strain) were prepared on *Ae. albopictus* C6/36 cells. For quantification of viral stocks, SINV and CHIKV were titrated on BHK-21 cells and DENV2 was titrated on BHK-15 cells.

To determine infectious DENV titres upon helicase silencing, end-point dilution assays were performed. A day prior to the titration, $1 \times 10^4$ BHK-15 cells per well were seeded in a 96-well flat bottom plate. A 10-fold serial dilution of virus samples was added to the cells in quadruplicate. After 7 days incubation, cells were inspected for cytopathic effect (CPE). The virus titre was calculated according to the Reed and Muench method [37].

## *Ae. aegypti* mosquito rearing, dissection, and intrathoracic injection

*Ae. aegypti* mosquitoes (Black Eye Liverpool strain, obtained from BEI resources) were reared at 28°C and 70% humidity with automated room lighting set at a 12:12 hours light/dark cycle.

Larvae were fed with Tetramin Baby fish food (Tetra). Adult mosquitoes were fed with a 10% sucrose solution. Three to five-day-old female mosquitoes (n = 30) were dissected as previously described [38]. Entire mosquitoes or dissected tissues (ovaries, midgut, head, thorax, and rest of the body) were homogenized in 300 µl RNA-Solv reagent (Omega Bio-Tek) using a Precellys 24 homogenizer (Bertin technologies). To the homogenates, 700 µl RNA-Solv reagent was added and total RNA was isolated according to manufacturer's recommendation (briefly described below).

To assess the effect of *Dhx15* silencing *in vivo*, three to five-day-old female mosquitoes were anesthetized on ice and intrathoracically injected with 1 µg dsRNA dissolved in 250 nl RNase-free water. Two days later, surviving mosquitoes were anesthetized with $CO_2$ and intrathoracically injected with 50 nl Leibovitz L-15 medium containing 150 plaque-forming units (PFU) CHIKV. Two days later, living mosquitoes were selected and homogenized in 1 ml RNA-Solv reagent for quantification of virus levels and gene expression.

## Cloning of expression constructs

cDNAs of Dhx15, AAEL004859, and AAEL008728, were cloned into pUbGw and pU3Fw for N-terminal tagging with GFP or 3xFlag, respectively. The vector pUbGw was modified from the expression vector pUbB-GW, (kindly provided by Dr. ir. Gorben Pijlman, University of Wageningen), as previously described [39]. The expression vector pU3Fw was derived from the pUbGw vector by exchanging the GFP sequence with a 3xFlag tag [40]. For AAEL004859 and AAEL008728, gene-specific primers were used to amplify the genes from Aag2 cDNA, and PCR products were inserted into an intermediate cloning vector using the TOPO-TA cloning kit (Thermo Fisher) according to the manufacturer's protocol. The obtained plasmids were then used as PCR template for In-Fusion HD cloning (Takara). Purified PCR products were inserted into the Gateway entry vector pDonor/Zeo (Invitrogen) using the In-fusion reaction according to the manufacturer's protocol. The sequence of the entry vector was confirmed by Sanger sequencing and subsequently, LR-recombination (Thermo Fisher) was performed to recombine the sequence of the genes of interest to the destination vectors pUbGw and pU3Fw. For Dhx15, PCR amplification with Gateway cloning compatible primers was performed directly on Aag2 cDNA using CloneAmp Hifi PCR pre-mix (Takara), without prior amplification in a TOPO TA cloning vector. The PCR product was inserted in the pDonor/Zeo entry vector and recombined into the destination vectors using the Gateway cloning protocol (Thermo Fisher) as described above. Primer sequences used for cloning are provided in S2 Table.

## Double-stranded RNA production

dsRNA targeting the 461 selected RNA-binding proteins or Argonaute-2 (Ago-2) and firefly luciferase as positive and negative controls, respectively, were produced from T7 promoter-flanked PCR products. The T7 sequence was either directly present in the primer sequence used to generate the PCR products or it was introduced during a second PCR step using T7 universal primers that hybridize to short GC-rich tags that were introduced to the PCR products in the first PCR (see S2 Table for primer sequences). These PCR products were *in vitro*-transcribed using an in-house T7 polymerase enzyme. For the formation of dsRNA, the reaction products were heated to 90°C for 10 minutes and then allowed to gradually cool down to room temperature. To purify dsRNA, GenElute Mammalian Total RNA (Sigma) or GenElute 96 Well Total RNA purification kits (Sigma) were used according to the manufacturer's protocols.

## Transfection and infection of Aag2 cells

For knockdown experiments, Aag2 cells were seeded at a density of $1.5 \times 10^5$ cells/well in a 24-wells plate or $5 \times 10^4$ cells/well in a 96-wells flat bottom opaque white plate. For each condition, 3 wells were seeded 24 hours prior to the first dsRNA transfection. In the 24-wells plate format, transfection mixes containing 300 µl non-supplemented L-15 medium, 450 ng dsRNA, and 1.8 µl X-tremeGENE HP DNA transfection reagent (Roche) were prepared according to the manufacturer's instructions. Per well, 100 µl of the transfection mix was added in a dropwise manner. For the 96-wells plate format, the volumes and amounts of transfection mix components was one third of the quantities used for 24-wells plates. Three hours post-transfection, the medium was replaced with supplemented L-15 medium. To enhance knockdown efficiency, the transfection was repeated 48 hours after the first transfection. Aag2 cells were virus infected at the indicated multiplicity of infection (MOI) when changing the medium after the second transfection and cells were harvested 48 hours post-infection for downstream analyses.

For assessing virus replication after *Dhx15* transgene expression, Aag2 cells were seeded in a 24-wells plate and transfected with pUbGw-Dhx15 and the empty pUbGw vector when reaching 70–80% confluency. 1.5 µg plasmid DNA and 3 µl X-tremeGENE HP DNA transfection reagent were mixed in 300 µl non-supplemented L-15 medium and after 20 min incubation, 100 µl of the transfection mix was added per well. Four hours later, the medium was refreshed with supplemented L-15 medium. Cells were infected with SINV-nluc 24 hours post-transfection and luciferase assays were performed 48 hours post-infection.

## Cell fractionation

Aag2 cells were seeded 24 hours prior to plasmid transfection at a density of $3.7 \times 10^6$ cells/well in a 6-well plate. For each reaction, transfection mixes were prepared containing 500 µl non-supplemented L-15 medium, 5 µg plasmid DNA (Flag-tagged helicases), and 5 µl X-tremeGENE HP DNA transfection reagent. Where indicated, cells were infected with SINV after the transfection and samples were harvested 48 hours post-infection. For sample preparation, Aag2 cells were resuspended, washed with PBS, and pelleted at 300 x *g* for 5 min. Next, cell pellets were lysed using cytoplasmic lysis buffer (50 mM NaCl, 25 mM Tris-HCl pH 7.5, 2 mM EDTA, 1x protease inhibitor, 0.5% NP40) and the cytoplasmic and the nuclear fractions were separated after 10 minutes centrifugation at 9600 x *g* at 4°C. To the supernatant (cytoplasmic fraction) 5x Laemmli buffer (4% SDS, 0.004% bromophenol blue, 0.125 M Tris-HCl pH 6.8, 20% glycerol, 10% 2-mercaptoethanol) was added to a final concentration of 1x; the nuclear pellet was resuspended in Laemmli buffer diluted to 1x in cytoplasmic lysis buffer. For the western blot analysis, lysate fractions representing equal number of cells were loaded on gel.

## Co-immunoprecipitation

For co-transfection, $2.2 \times 10^7$ Aag2 cells were seeded in a T-75 flask and transfected with 30 µg of each plasmid (GFP- and Flag-tagged helicases) using 60 µl X-tremeGENE HP DNA transfection reagent. After two and a half hours incubation at 28°C, the transfection medium was replaced with supplemented L-15 medium.

Aag2 cells co-expressing GFP- and Flag-tagged RNA helicases were lysed in RIPA buffer (1% Triton X-100, 150 mM NaCl, 0.1% SDS, 0.5% Na-deoxychelate, 50 mM Tris pH 8.0, 1x protease inhibitor). The lysate was subjected to affinity enrichment using magnetic GFP-TRAP beads (ChromoTek) following the manufacturer's protocol. Briefly, beads were washed in washing buffer (10 mM Tris-HCl pH 7.5, 150 mM NaCl, 0.5 mM EDTA, 1x complete-EDTA free protease inhibitors, and 1 mM PMSF). Where indicated, the samples underwent RNase A (Thermo Fisher) treatment for 7.5 minutes at 37°C. After RNase A treatment, at least one

additional washing step was performed prior to the final elution. To the input samples, the samples of washing steps, and the final eluate, 5x Laemmli buffer was added. Samples were heated at 90°C for 10 minutes and analysed on western blot.

## Western blotting

For western blotting, protein samples were separated on polyacrylamide gels, blotted to nitrocellulose membranes and probed with the indicated antibodies. The primary antibodies used were mouse anti-H3K9me2 (Abcam ab1220), rat anti-α-tubulin (Sanbio), mouse anti-Flag M2 (Sigma), and rat anti-GFP (ChromoTek). The secondary antibodies used were IRdye680 or IRdye800 conjugated goat anti-rat or goat anti-mouse antibodies (LI-COR). All primary antibodies were diluted 1:1000, and secondary antibodies were diluted 1:10000. Western blots were imaged on an Odyssey CLX imaging system (LI-COR).

## RNA isolation

Aag2 cells were homogenized in RNA-Solv reagent (Omega Bio-Tek) and RNA extraction was performed as described in the manufacturer's instructions. Briefly, to 1 mL RNA-Solv reagent, 200 μl of chloroform was added and thoroughly mixed. After centrifugation, the aqueous phase was collected, and RNA was precipitated using one volume of isopropanol for 1 hour at 4°C followed by centrifugation to pellet the RNA. Pellets were washed twice in 80% ethanol, dissolved in nuclease free water, and RNA was quantified using a Nanodrop spectrophotometer.

## Reverse transcription and (quantitative) PCR

For reverse transcription followed by quantitative polymerase chain reaction (RT-qPCR), 1 μg of RNA was DNase I (Ambion) treated according to the manufacturer's protocol and reverse transcribed using the TaqMan MultiScribe Reverse Transcription Kit (Applied Biosystems) using poly-dT or random hexamer primers. Quantitative PCR was performed on a LightCycler 480 (Roche) using the GoTaq qPCR Mix (Promega), according to the manufacturer's protocol. Relative expression of target genes was calculated using the $2^{(-\Delta\Delta CT)}$ method [41] for which the expression of *lysosomal aspartic protease* (*LAP*) or *ribosomal protein L5* (*RpL5*) was used as an internal reference. End-point PCR to detect gene expression in mosquito tissues was performed using GoTaq polymerase (Promega) according to the manufacturer's instructions. Sequences of primers are indicated in S2 Table.

## Luminescence and cell viability assays

The *Renilla*-Glo Luciferase assay kit (Promega) was used to quantify nLuc reporter expression in all knockdown experiments. The buffer volume was reduced to 70 μl of the reconstituted *Renilla*-Glo luciferase reagent per well of a 96-well plate. To assess nluc reporter expression in the context of *Dhx15* transgene expression, the *Renilla* luciferase assay system (Promega) was used. Briefly, cells were lysed in 120 μl 1x *Renilla* luciferase assay lysis buffer and 10 μl cell lysate was mixed with 25 μl 1x *Renilla* luciferase assay substrate. The CellTiter-Glo 2.0 assay (Promega) was used to quantify viable cells, according to the manufacturer's instructions. Luminescence was measured on a Modulus single tube reader or Perkin Elmer Counter Victor 3 plate reader.

## RNA-sequencing library preparation and analysis

The TruSeq Stranded mRNA kit (Illumina) was used for single-end RNA sequencing library preparation from total RNA according to the manufacturer's protocol. The input for the

library preparation was 1 µg RNA to obtain double-stranded cDNA. The prepared libraries were quantified and controlled for sample quality using a DNA1000 Bioanalyzer (Agilent). The libraries were sequenced on an Illumina HiSeq 4000 platform (GenomEast Platform).

## Bioinformatics analyses

After initial quality control by the sequencing platform, raw sequence reads were aligned to the *Ae. aegypti* LVP_AGWG AaegL5.1 reference genome (retrieved from VectorBase) using STAR 2.5.0 [42] with default settings. A detailed summary of the RNA-sequencing data can be found in S3 Table. R package DESeq2 [43] using read count per gene was used for statistical analysis of differential gene expression (with adjusted *p*-value < 0.05) and principal-component analysis. Genes were considered expressed if the mean of the DESeq2-normalized counts (baseMean) was higher than 10. The R package pheatmap (RRID:SCR_016418) was used to generate the heatmap for differentially expressed genes upon CHIKV infection, which was based on z-scores of normalized gene expressions (log10FPKM). The heat maps showing differential expression of glycolytic genes (based on log2-transformed fold changes) were generated in Microsoft Excel using the three colour scale option of the conditional formatting function. Expression analysis of helicases in published datasets was performed as described previously [44,45]. Briefly, publicly available datasets were retrieved from NCBI Sequence Read Archive and mapped to the AaegL5 genome using STAR aligner version 2.5.2b [42]. Raw read counts were then normalized with DESeq2 [43] and plotted with ggplot2 [46]. GO term enrichment analysis was performed using DAVID (Database for Annotation, Visualization and Integrated Discovery) [47,48]. The STRING database was used to predict protein-protein interactions [49]. Domain structure of RNA helicases was retrieved from Simple Modular Architecture Research Tool (SMART) (http://smart.embl-heidelberg.de/).

Phylogenetic analysis of RNA helicases was performed using the Multiple sequence alignment tool available on GenomeNet operated by the Kyoto University Bioinformatics Center (https://www.genome.jp/tools-bin/clustalw). As input, the protein sequences of the DEAD domains of *D. melanogaster* and *Ae. aegypti* DEAD-box RNA helicases were used. For this analysis, DEAD-box helicases were identified using the 'search for' function in VectorBase asking for gene identifiers based on InterPro Domain database. The specific domain to be searched for was PF00270: DEAD DEAD/DEAH-box helicase. The resulting gene lists were obtained for both *D. melanogaster* and *Ae. aegypti* and the 'edit -orthologues' function was used to identify orthologous genes in the other species, respectively. The obtained lists were compiled into one non-redundant gene list of DEAD-box helicases for each species. Amino acid sequences of the DEAD domain of each protein were retrieved from the SMART database, or if unavailable, manually extracted from the protein sequences using the amino acid coordinates given by PFAM. The approximately maximum-likelihood tree was generated on the multiple sequence alignment using the FastTree full algorithm in GenomeNet, based on FastTree 2 [50].

## 2-deoxy-D-glucose treatment and lactate concentration measurement

Aag2 and Hela cells were seeded 24 hours prior to 2-deoxy-D-glucose (Sigma) treatment. Cells were incubated for 48 hours with either 24 mM or 50 mM 2-deoxy-D-glucose, harvested, and samples were analysed using a lactate assay kit (Sigma-Aldrich). Lactate concentration was measured using a colorimetric detection following the manufacturer's instructions.

## Statistical analysis and data availability

Unless indicated differently, experiments had three biological replicates and the data are represented as mean and standard deviation. Statistical significance was attributed when *p*-values

were <0.05. Graphs were generated and statistical analyses were performed using GraphPad Prism (version 8.0.0 for Windows) or IBM SPSS v25. Unprocessed numerical data is available in S4 Table. RNA-sequencing data is deposited under the BioProject accession PRJNA885496.

## Results

### A targeted RNAi screen in mosquito cells identifies novel host genes that control arbovirus replication

To identify RBPs that control virus replication in *Ae. aegypti*, we designed a targeted knock-down screen in Aag2 cells. Using the biomart plugin in VectorBase (release 2017–8), we selected all genes from the *Ae. aegypt*i L3.3 genome annotation that were associated with the GO term 'RNA binding'. We also identified *Ae. aegypti* orthologues of predicted RBPs in three additional mosquito species (*Ae. albopictus*, *Culex quinquefasciatus*, and *Anopheles gambiae*) as well as the fruit fly *Drosophila melanogaster* and combined all datasets into a non-redundant list of 635 genes. We manually excluded 132 genes that were part of the core transcription, splicing, and translation machineries. Another 42 genes were omitted because the PCR ampli-fication to generate the template for *in vitro* transcription of dsRNA repeatedly failed. Overall, we managed to successfully produce dsRNA for knockdown of 461 genes, which represented the set of genes included in the first screening round (S1 Table).

All genes were individually silenced in Aag2 cells followed by infection with a SINV infec-tious clone expressing a nano-luciferase reporter gene as a fusion protein with nsP3 [35] (Fig 1A). In the initial screening round, knockdown of 38 and 49 genes resulted in a ≥ 2-fold increase or decrease of luciferase levels, respectively, compared to a non-targeting control knockdown (Fig 1B and 1C). We repeated the knockdown experiment for these genes using the same dsRNA preparation and, for those that showed a reproducible phenotype, we gener-ated a second set of dsRNA targeting a different region of the transcript to account for possible off-target effects (Fig 1D, S1 Table). As controls, we included silencing of the antiviral RNAi core factor *Ago2* and knockdown of the SINV genomic RNA itself. With this extensive confir-mation procedure, we validated the phenotype of fifteen antiviral (Fig 1D) and four proviral hits (S1 Table).

Here, we focused on the genes that enhanced virus replication upon knockdown, as those are putative players in antiviral defense. Importantly, knockdown of these genes did not, or only mildly, affect cell viability (S1A Fig). To validate the antiviral activity using an indepen-dent readout, we assessed the effect of gene knockdown on SINV replication at the RNA level. Efficient silencing could be verified for most genes (S1B Fig) and, analogous to our findings measuring luciferase, resulted in an increase of viral RNA levels (Fig 1E), underscoring the robustness of our screening approach.

Amongst the hits of our RNAi screen, we identified five predicted DEAD-box RNA heli-cases (AAEL001216, AAEL004419, AAEL004859, AAEL006794, and AAEL008728) amongst which the known antiviral RNAi factor Dicer 2 (AAEL006794) and four RNA helicases that had not previously been associated with antiviral activity in mosquitoes. Besides the five DEAD-box RNA helicases, we identified four proteins with predicted functions in transcrip-tional regulation (AAEL006956, AAEL006566, AAEL002125, and AAEL026633) and miscella-neous proteins predicted to be involved in (pre-) mRNA metabolism, export, or splicing (AAEL009061, AAEL001007, and AAEL0011663). One hit was a predicted regulator of cell cycle (AAEL000707) and two hits were uncharacterized proteins (AAEL020123 and AAEL010312).

Given the importance of DEAD-box RNA helicases in modulating immune signaling, we further focused our analysis on the uncharacterized RNA helicases. First, we aimed to establish

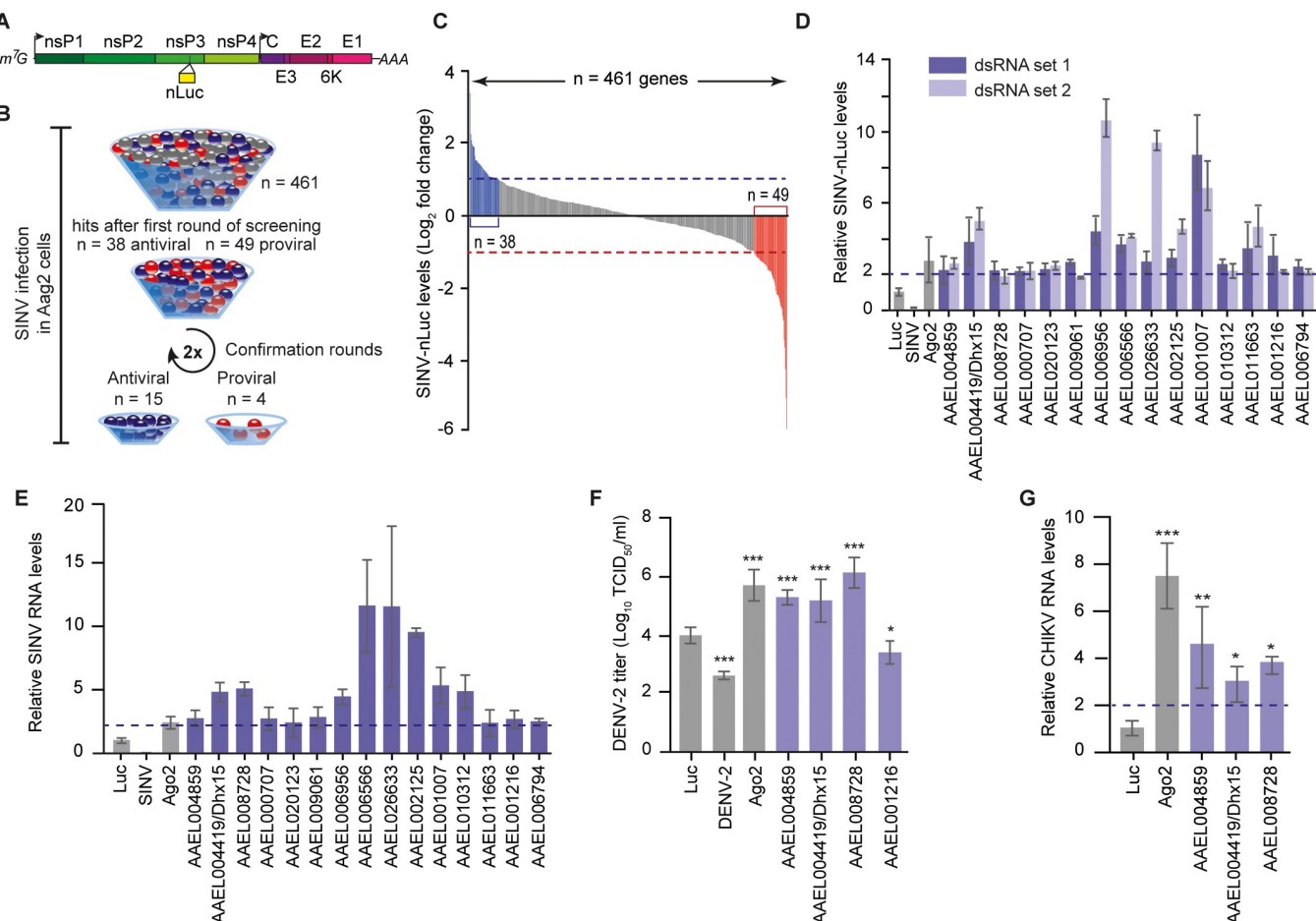

**Fig 1. A targeted RNAi screen identifies RNA-binding proteins (RBPs) that control arboviruses replication in mosquito cells. A)** Schematic representation of recombinant SINV infectious clone expressing a nano-luciferase reporter gene as a fusion protein with nsP3. The individual non-structural and structural viral proteins are depicted in different shades of green and purple, respectively. The position of the nLuc is marked by the yellow bar. **B)** Schematic flow of the RNAi screen. Antiviral, proviral, and neutral genes were depicted in blue, red, and gray, respectively. **C)** SINV-nluc levels, measured by luminescence, upon individual silencing of 461 genes in Aag2 cells. The 2-fold threshold (note that data is plotted on $\log_2$ scale) is indicated with dashed lines and putative antiviral and proviral genes are depicted in blue and red, respectively. SINV-nluc infection was performed with MOI = 0.1. Bars are means of three replicates. **D)** Validation of the RNAi screen. Candidate genes were silenced in Aag2 cells using two independent sets of dsRNA and virus replication was measured with a luminescence assay. The 2-fold increase on the linear scale is indicated by the dashed line. **E)** Quantification of SINV RNA levels by RT-qPCR after silencing of the indicated genes in Aag2 cells using the first set of dsRNA. The 2-fold increase on the linear scale is indicated by the dashed line. **F)** Infectious DENV-2 titers (note that data is plotted on $\log_{10}$ scale) in the supernatant of Aag2 cells upon *AAEL004419/Dhx15*, *AAEL008728*, *AAEL004859*, and *AAEL001216* silencing. DENV-2 infection was performed at an MOI of 0.1. **G)** Quantification of CHIKV RNA levels by RT-qPCR after silencing of *AAEL004419/Dhx15*, *AAEL008728*, and *AAEL004859*. CHIKV infection in Aag2 C3PC12 cells was performed with MOI = 0.1. The 2-fold increase on the linear scale is indicated by the dashed line. In panels (***D-G***), bars and whiskers represent the mean +/- SD of three independent biological replicates. In (***F***) and (***G***), statistical significance was determined using One-Way ANOVA with Holm-Sidak correction (* $p < 0.05$, ** $p < 0.005$, *** $p < 0.0005$).

the antiviral activity of these DEAD-box helicases against other arboviruses. Silencing of *AAEL004419*, *AAEL008728*, and *AAEL004859*, but not *AAEL001216* resulted in a profound increase of dengue virus titers, to similar levels as silencing of *Ago2* (Fig 1F). Similarly, knockdown of *AAEL004419*, *AAEL008728*, and *AAEL004859* in an Aag2-derived clonal cell line (Aag2 C3PC12) that has been cleared from persistent virus infections [34] enhanced RNA replication of CHIKV by > 2-fold (Fig 1G), suggesting a broad antiviral activity of these helicases. Importantly, silencing of the identified RNA helicases in Aag2 C3PC12 cells also resulted in increased SINV levels as observed in the initial knockdown screen, which had been performed in the parental Aag2 cell line (S1C and S1D Fig).

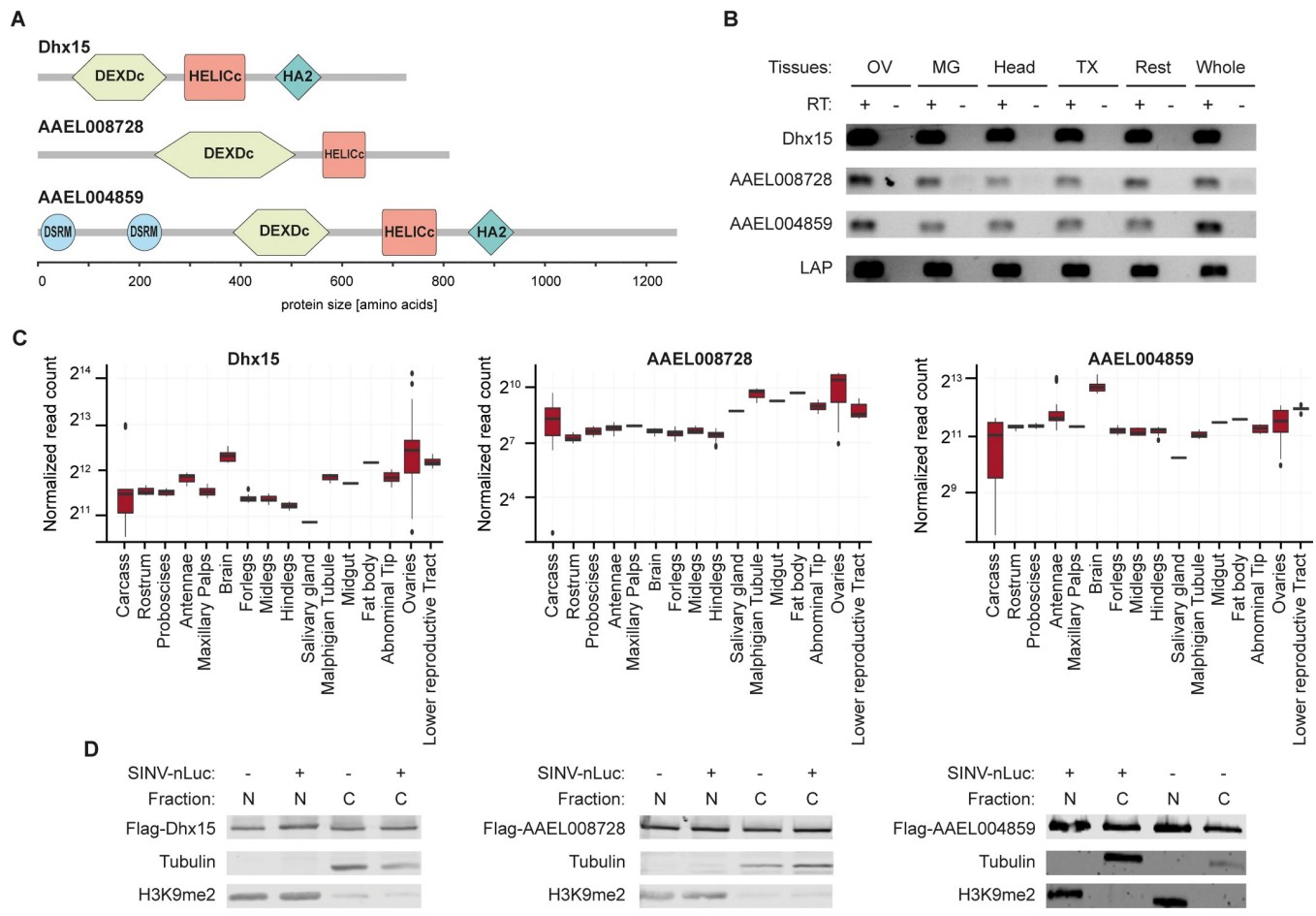

**Fig 2. Characterization of AAEL004419/Dhx15, AAEL008728, and AAEL004859. A)** Schematic representation of the domain composition of the RNA helicases AAEL004419/Dhx15, AAEL008728, and AAEL004859 predicted with SMART. DEDXc: DEAD-like helicase superfamily domain, HELICc: helicase superfamily C-terminal domain, HA2: C-terminal helicase associated domain, and DSRM: dsRNA binding motif. **B)** Expression of *AAEL004419/Dhx15*, *AAEL008728*, *AAEL004859*, and the house-keeping gene *Lysosomal Aspartic protease* (*LAP*) assessed by RT-PCR on ovaries (OV), midgut (MG), head, thorax (TX), and rest of the body dissected from female *Ae. aegypti* mosquitoes as well as in entire mosquitoes. PCR amplification on samples without reverse transcriptase (RT -) served as negative control. **C)** Expression of *AAEL004419/Dhx15*, *AAEL008728*, and *AAEL004859* in mosquito tissues in published RNA-sequencing datasets. **D)** Cellular localization of the proteins of interest in noninfected (-) and SINV infected (+) Aag2 C3PC12 cells. SINV-nLuc infection was performed at MOI = 0.1. Cell fractionation assays followed by western blot show the expression of AAEL004419/Dhx15, AAEL008728, and AAEL004859 in the nucleus (N) and in the cytoplasm (C).

## Characterization of broadly antiviral DEAD-box RNA helicases

AAEL004419, AAEL008728, and AAEL004859 are canonical DEAD-box helicases containing DEAD-like helicase superfamily (DEXDc) and helicase superfamily C-terminal (HELICc) domains. In addition, AAEL004419 and AAEL004859 contain a C-terminal helicase associated (HA2) domain and AAEL004859 contains two dsRNA binding motifs (DSRM) (Fig 2A). Alignment of *Ae. aegypti* and *Drosophila* DEAD-box helicase domains, identified Dhx15, CG9143, and maleless (mle) as the closest orthologs of AAEL004419, AAEL008728, and AAEL004859, respectively (S2A Fig). In particular, AAEL004419 is highly conserved with about 90% amino acid identity across all functional domains (S2B Fig). Because of this close one-to-one orthology, we will refer to AAEL004419 as Dhx15.

To further characterize the three DEAD-box helicases, we investigated their expression pattern both at the tissue level in adult mosquitoes and on subcellular level in Aag2 cells. In dissected female *Ae. aegypti* mosquitoes, we found Dhx15, AAEL008728, and AAEL004859 to be

ubiquitously expressed across all somatic and germline tissues analyzed (Fig 2B), which is in line with published RNA expression data (Fig 2C). To assess the subcellular localization of Dhx15, AAEL008728, and AAEL004859, we expressed Flag-tagged proteins in Aag2 cells and performed nuclear versus cytoplasmatic fractionation. Efficient separation of the cytoplasmic and nuclear fractions was confirmed by the segregation of tubulin and Histone H3 lysine 9 dimethylation (H3K9me2) markers, respectively. We identified all three RNA helicases to be ubiquitously expressed in the nuclear and in the cytoplasmic fractions both in uninfected and SINV infected Aag2 cells, indicating that distribution across these cellular compartments was not altered as a response to virus infection (Fig 2D).

## Silencing of *Dhx15* results in an altered transcriptional response regulating glycolysis

Transgenic expression of *Dhx15* decreased SINV nluc levels in Aag2 cells, providing additional support for the antiviral phenotype of this helicase in Aag2 cells (S2C Fig). Interestingly, in vertebrates, the orthologues of Dhx15 have previously been proposed to regulate transcriptional responses to virus infection by modulating signal transduction of core immune pathways such as MAPK (mitogen-activated protein kinase), NFκB, and RIG-I like receptor (RLR) signaling [51–53]. We therefore assessed transcriptional regulation mediated by the highly conserved RNA helicase Dhx15 in mosquito cells. After sequential knockdown of *Dhx15* in Aag2 C3PC12 cells, we performed RNA-sequencing and gene expression analysis (Fig 3A). Genes were considered differentially expressed (DE) when their expression levels were up or downregulated by at least 2-fold and the adjusted *p*-value was $p < 0.05$. Using these parameters, we identified 528 genes upregulated and 328 genes downregulated upon *Dhx15* knockdown (Fig 3B). For the upregulated genes, GO terms related to DNA replication were the most strongly enriched; for the downregulated genes, GO terms related to sugar metabolism, most prominently *glycolytic process*, were the most strongly enriched (Fig 3C). These results indicated that Dhx15 directly or indirectly controls a transcriptional response in Aag2 cells.

We next assessed the effect of *Dhx15* knockdown on gene expression in the context of virus infection. Aag2 cells were infected with CHIKV shortly after the second knockdown, and RNA samples were taken 48 hours post-infection (Fig 3A). Efficient CHIKV replication and *Dhx15* knockdown were verified in these samples (S3A and S3B Fig) and analysis of RNA deep-sequencing data identified 229 genes and 143 genes to be significantly up or downregulated, respectively (Fig 3D). The majority of these (194 out of 229 upregulated genes and 89 out of 143 downregulated genes) overlapped with the differentially expressed genes in uninfected samples, defining a set of genes with robust *Dhx15*-dependent differential expression, regardless of virus infection (S3C Fig). Interestingly, while for the upregulated genes, *DNA templated regulation of transcription* was the only enriched GO term, for the downregulated genes, GO terms were highly concordant between uninfected and infected conditions with *glycolytic process* being the most strongly enriched (Fig 3E). We therefore specifically analyzed the expression of genes that are part of the glycolysis pathway, and indeed found that the entire set of glycolytic core enzymes was downregulated upon *Dhx15* knockdown, in particular those that are involved in the metabolic conversion of glucose to glyceraldohyde-3-phosphate (Figs 3F and S3D).

Amongst the most strongly downregulated genes was the gene encoding phosphofructokinase, the enzyme that performs the rate-limiting step of the glycolysis pathway. We therefore functionally assessed the effect of *Dhx15* knockdown on the glycolytic rate by measuring the concentration of lactate, a fermentation product of the glycolysis pathway as a proxy for activity [54–56]. To benchmark our assay, we treated Aag2 cells with 2-deoxy-D-glucose (2-DG),

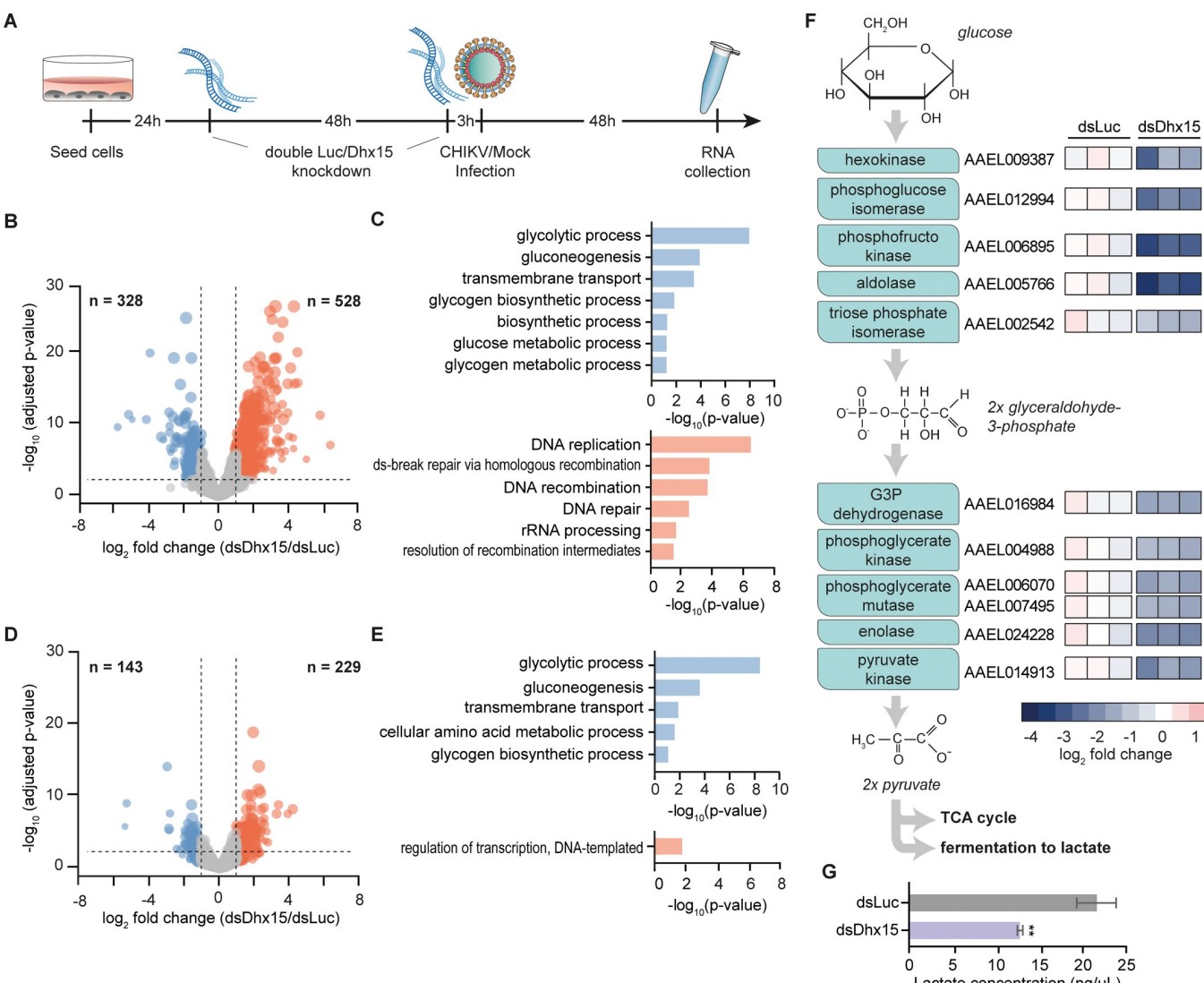

**Fig 3. Dhx15 regulates a transcriptional response that controls glycolysis. A)** Set-up of RNA-sequencing analysis to assess the transcriptomic response to *Dhx15* silencing. 24 hours and 72 hours after Aag2 C3PC12 cells were seeded, a sequential knockdown of *Dhx15* (dsDhx15) or a non-targeting Firefly luciferase (dsLuc) control was performed. CHIKV (MOI = 5) or mock infection was performed 3 hours after the second knockdown and samples were collected 48 hours later. **B)** Volcano plot of differentially expressed genes upon *Dhx15* silencing. Downregulated genes and upregulated genes are indicated in blue and red, respectively. The X-axis denotes log2 fold change values; the Y-axis shows -log10 (*p*-value). **C)** GO terms of differentially expressed genes upon *Dhx15* silencing. The upper panel (blue) shows the GO analysis of downregulated genes, the lower panel (red) indicates the GO annotation of upregulated genes. **D)** Volcano plot of differential expressed genes upon *Dhx15* silencing in the context of CHIKV infection, showing downregulated genes and upregulated genes in blue and red, respectively. The X-axis denotes log2 fold change values; the Y-axis shows -log10 (*p*-value). **E)** GO annotation of differentially expressed genes upon *Dhx15* silencing in CHIKV infected cells as in panel *(C)*. **F)** Schematic representation of the enzymes involved in the glycolysis pathway (left) and log2 fold changes of these genes upon *Dhx15* or control silencing (right). **G)** Relative lactate concentration upon *Dhx15* or luciferase silencing in Aag2 C3PC12 cells. Bars and whiskers represent the mean +/- SD of three independent biological replicates. Statistical significance was determined using unpaired two tailed t-test (** *p* < 0.005).

which is converted by hexokinase into 2-deoxy-D-glucose phosphate, a competitive inhibitor of phosphoglucose isomerase at the second step of glycolysis [57]. As a control, we treated Hela cells, for which 2-DG treatment is known to reduce lactate concentration [58]. As expected, treatment with 2-DG resulted in an almost 30% decline of lactate levels in Hela cells (S3E Fig). In contrast, in Aag2 cells, baseline lactate levels were lower and treatment with 2-DG only had a minor effect on lactate concentration (S3F Fig). We hypothesized that this

may be explained by the composition of the L-15 culture medium, which contains galactose instead of glucose and additional high levels of pyruvate. Galactose can enter glycolysis but at lower efficiency than glucose and high levels of pyruvate favor energy production by directly entering the TCA cycle, which likely reduces the glycolytic activity to form lactate. To sensitize the lactate assay, we therefore cultured Aag2 cells in Schneider's medium, which is supplemented with glucose and does not contain pyruvate. Importantly, also in these culture conditions, CHIKV replication was efficient and reached similar endpoint RNA levels as in cells grown in L-15 medium, although we noticed that replication was accelerated at early timepoint after infection (S3G Fig). In Aag2 cells cultured in Schneider's medium, baseline lactate levels were elevated, and 2-DG treatment resulted in significantly lower lactate concentrations, indicating that we were able to measure alterations in glycolytic activity in Aag2 cells (S3F Fig). We next assessed lactate levels upon *Dhx15* silencing. Strikingly, we observed a profound decrease of lactate concentration in cell homogenates, even exceeding the effect of 2-DG treatment, indicating that the reduced expression of glycolysis genes upon *Dhx15* knockdown resulted in a functional reduction of glycolytic activity in Aag2 cells (Fig 3G). To exclude that the decrease in lactate concentration was caused by a decline in cell number, we counted cells directly before performing the lactate assay. Cell counts remained stable upon *Dhx15* knockdown (S3H Fig) indicating that silencing of this helicase effectively reduces the intracellular lactate concentration.

## Transcriptional control of glycolytic genes is specific to Dhx15

We next aimed to investigate whether, besides Dhx15, the other identified antiviral DEAD-box helicases contributed to transcriptional downregulation of glycolytic genes. This hypothesis was sparked by a protein-protein interaction map that we generated for all 15 antiviral hits picked up in our screen using the STRING algorithm. In this analysis, all identified DEAD-box helicases were predicted to interact in a protein complex (Fig 4A). To confirm a direct protein-protein interaction of AAEL008728 with Dhx15 and AAEL004859 experimentally, we performed co-immunoprecipitations (Co-IP) in Aag2 cells. Confirming the predicted protein interaction network, Flag-tagged AAEL008728 was efficiently co-precipitated both by GFP-tagged AAEL004859 and Dhx15 (Fig 4B). Since the three DEAD-box helicases are predicted to have RNA binding activity, we next tested if their interaction was mediated indirectly through binding to the same RNA molecules. Therefore, we performed Co-IP in the presence of RNase A to disrupt RNA-bridged protein-protein interactions. Dhx15 and AAEL008728 binding was resistant to RNase A treatment (Fig 4C), indicating an RNA-independent interaction between these helicases.

Having confirmed a direct interaction of the identified DEAD-box helicases, we next assessed if knockdown of *AAEL008728* and *AAEL004859* caused a similar transcriptional response as Dhx15. Quantification of glycolytic genes and an additional selection of differentially regulated genes from the RNA-sequencing data confirmed reduced gene expression upon silencing of *Dhx15*. Yet, knockdown of *AAEL008728* and *AAEL004859* did not reduce expression of these genes (Figs 4D and S4A). These data indicate that the transcriptional control of glycolysis genes is independent from the identified protein complex of Dhx15, AAEL004859, and AAEL008728 (Fig 4A) but is mediated by Dhx15 in a separate molecular context.

## CHIKV infection downregulates glycolysis genes, akin to *Dhx15* knockdown

In response to virus infections, the activity of metabolic pathways is often changed reflecting the altered energy and biomolecule demand in infected cells [33]. Therefore, we wanted to

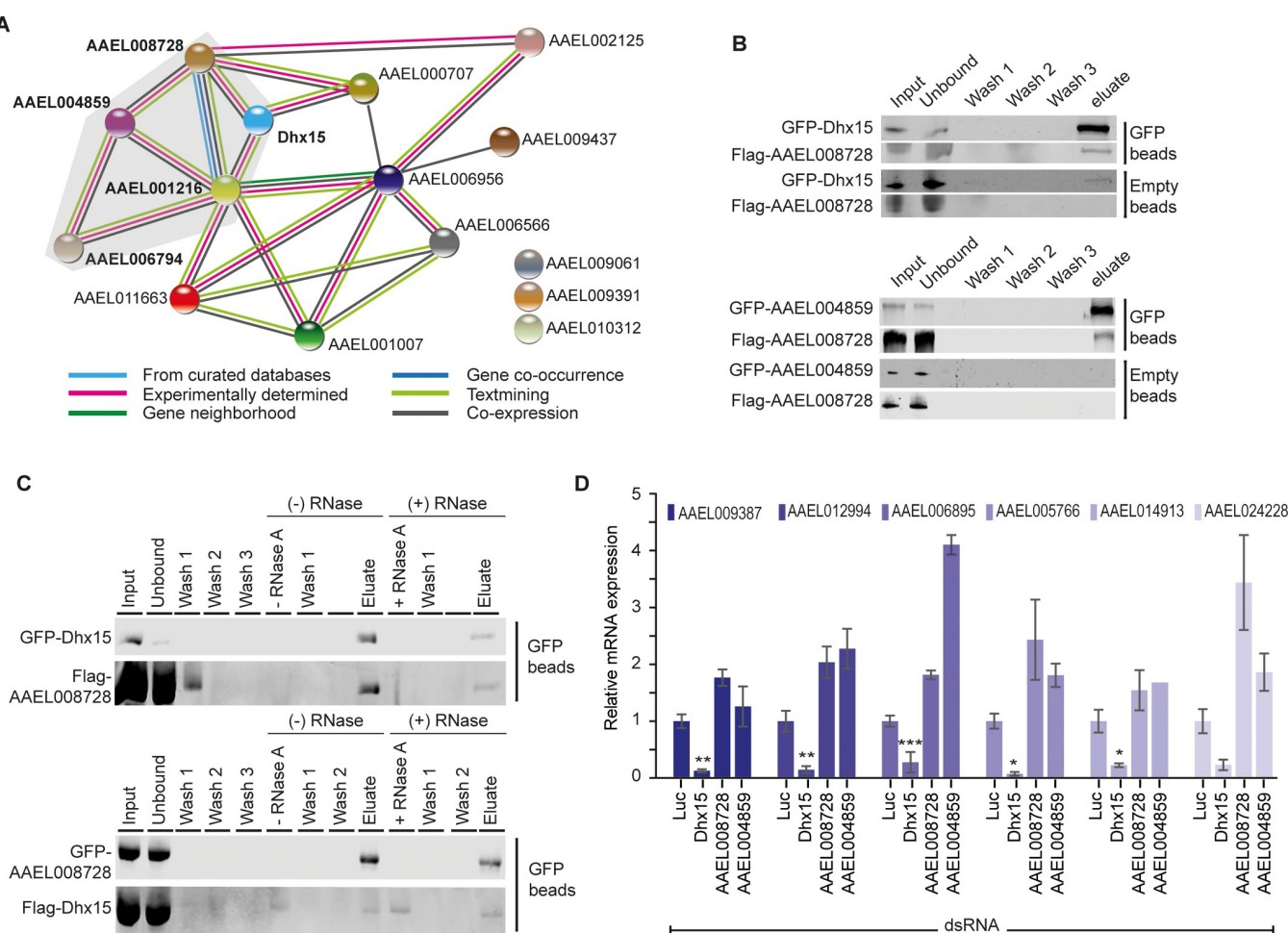

**Fig 4. Dhx15 downregulates glycolytic genes independently of interacting RNA helicases. A)** Protein-protein interactions predicted for fifteen antiviral RBPs using STRING. The color code of the lines connecting the different RBPs classifies the prediction for the protein-protein association. DEAD-box RNA helicases are highlighted with gray background. **B)** Western blot analysis of protein lysates from Aag2 cells transfected with GFP-Dhx15 and Flag-AAEL008728 (top panel) as well as GFP-AAEL004859 and Flag-AAEL008728 (bottom panel). Samples before (input) and after GFP-IP or control IP with empty beads were analyzed for co-purification of GFP- and Flag-tagged transgenes. Samples were probed with antibodies against GFP and Flag. **C)** Co-IPs of GFP-Dhx15 and Flag-AAEL008728 (top panel) and GFP-AAEL008728 and Flag-Dhx15 (bottom panel) from Aag2 C3PC12 cell lysate with (+) and without (-) subsequent on-bead RNase A treatment. RNase A was added to the sample after the 3 initial washing steps post-IP and samples taken directly after incubation are denoted as - RNase A and + RNase A in vertical writing, respectively. Samples were probed with antibodies against GFP and Flag. **D)** Quantification of glycolytic genes by RT-qPCR after silencing of *Dhx15*, *AAEL008728*, and *AAEL004859* in Aag2 C3PC12 cells. Bars and whiskers represent the mean +/- SD of three independent biological replicates. Statistical significance was determined using One-Way ANOVA with Holm-Sidak correction (* $p < 0.05$, ** $p < 0.005$, *** $p < 0.0005$).

assess the general transcriptional response of Aag2 cells to CHIKV infection. For this aim, we re-analyzed our RNA-sequencing data comparing gene expression in uninfected and CHIKV infected Aag2 cells using the samples that had been treated with non-targeting control dsRNA. This analysis allowed us to identify genes that were differentially regulated in response to CHIKV infection in Aag2 cells irrespective of *Dhx15* knockdown. In general, the transcriptional response to CHIKV in Aag2 cells was modest with only eight genes upregulated and 51 genes downregulated (Fig 5A). Strikingly, amongst the 51 downregulated genes, a significant number of genes (n = 22; Pearson Chi-square < 0.001) were also consistently decreased by *Dhx15* knockdown (Fig 5B), suggesting that CHIKV infection and *Dhx15* knockdown result in a partially overlapping transcriptional response. CHIKV-induced gene repression was not

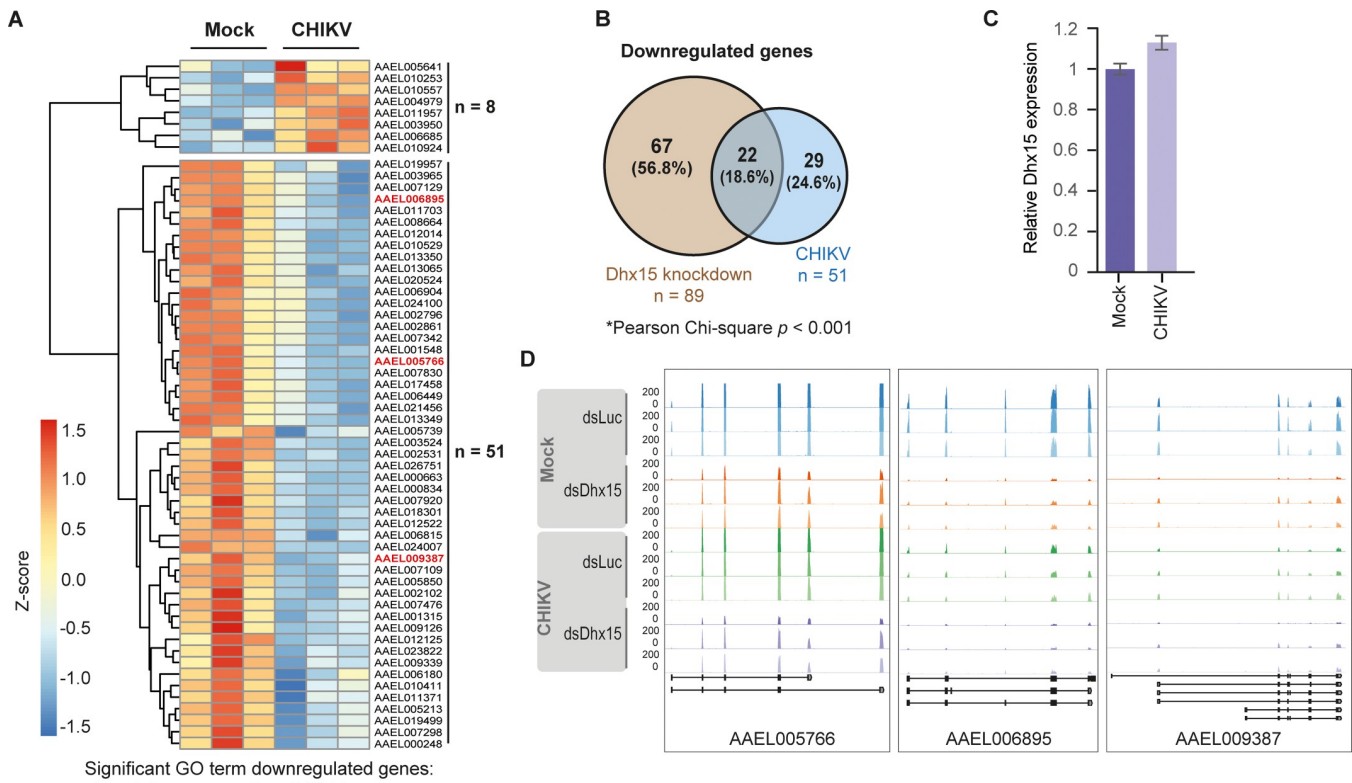

**Fig 5. Glycolytic genes are downregulated upon CHIKV infection. A)** Heatmap of differentially expressed genes upon CHIKV infection with an MOI of 5 (fold change $\geq$ 2; $p$-value < 0.05). Z-scores were calculated based on log10 fold changes of each gene to indicate the level of expression. Shades of red and blue indicate increased and decreased gene expression, respectively. **B)** Overlap of genes downregulated by *Dhx15* silencing and CHIKV infection as identified by RNA-sequencing. Pearson Chi-square test was used to test the zero hypothesis that the number of downregulated genes (n = 22) shared in both conditions is expected by chance. The zero hypothesis was rejected with $p$ < 0.001. **C)** Relative expression of *Dhx15* in uninfected and CHIKV infected cells, extracted from RNA-sequencing data. **D)** RNA-sequencing tracks for *AAEL05766* (*aldolase*), *AAEL006895* (*phosphofructokinase*), and *AAEL009387* (*hexokinase*) from the indicated conditions.

mediated by downregulation of *Dhx15*, as expression of this RNA helicase was not altered in infected cells (Fig 5C). Strikingly, GO analysis identified *glycolytic process* as the only enriched term (Fig 5A). The three core enzymes of the glycolysis pathway, aldolase, hexokinase, and the rate-limiting phosphofructokinase were significantly downregulated (Fig 5A and 5D). More general, all eleven glycolytic genes were expressed at a lower level in infected cells, albeit not always reaching our thresholds for minimal fold changes or statistical significance (S5A Fig). Particularly, the expression of enzymes involved in the first metabolic steps of glycolysis were reduced, akin to the effect of *Dhx15* knockdown (Figs 3F and S5A). Overall, these data indicate that both *Dhx15* knockdown and CHIKV infection lead to partly overlapping expression changes of glycolytic genes.

## Discussion

*Ae. aegypti* mosquitoes are important biological vectors for major arboviruses that impose a growing threat to human health [1], asking for a better understanding of the mechanisms that control virus growth in mosquitoes. Similar to other insect species, antiviral immunity in mosquitoes is governed by small RNA-mediated silencing of viral RNA as well as transcriptional responses to virus infections [11]. While the antiviral mechanisms underlying small RNA pathways, in particular RNAi, are relatively well established, mechanistic insights into how

transcriptional responses control antiviral immunity are limited [11,16,17], and additional, yet unknown, pathways that control virus replication in mosquitoes likely exist. In order to identify new players in antiviral defense, we performed a targeted knockdown screen in Aag2 mosquito cells, a cell line of embryonic origin that is frequently used to molecularly dissect antiviral immune pathways [21,59,60]. We focused this functional screen on RBPs, a protein family with pleiotropic functions in regulating immune responses across all domains of life [23–27]. Indeed, using a robust screening strategy, we identified several proteins with antiviral properties that, when silenced, resulted in increased virus replication. Amongst the hits were the well-established antiviral RNAi factor Dicer 2 (AAEL006794) and proteins that act in transcriptional pausing (Spt4: AAEL006566 and Spt6: AAEL006956), a process that has previously been reported to have antiviral activity in flies and Aag2 mosquito cells [61]. From the identified hits, we initially focused on DEAD-box RNA helicases and in particular on the role of Dhx15. Dhx15 exhibited broad antiviral activity against three RNA viruses: SINV and CHIKV from the alphavirus genus and DENV, a flavivirus. We showed that *Dhx15* controlled a transcriptional response that decreased glycolytic gene expression and activity in mosquito cells. Intriguingly, CHIKV infections resulted in a similar downregulation of genes involved in the glycolysis pathway. Although further mechanistic experiments are needed, our data suggest that the enhanced virus replication of CHIKV upon *Dhx15* knockdown may be explained by establishment of a metabolic environment that favors CHIKV replication. Thus, while our RNAi screen was initially intended to discover new immune factors that directly interfere with virus replication, we eventually identified a host protein that indirectly represses virus growth, potentially by controlling the metabolic state of the cell.

DEAD-box helicases have various functions in general RNA metabolism as well as controlling antiviral immunity [62–64]. Prime examples are Dicer 2 and RIG-I like RNA helicases, which are essential for sensing viral RNA in invertebrates and vertebrates, respectively [24,29]. In addition, other DEAD-box helicases modulate immune signaling via direct interaction with core signaling intermediates in the cytoplasm or by regulating transcription in the nucleus as co-activators or co-suppressors of transcription factors [65, 66]. As such, several DEAD-box helicases in mammals (for instance DDX1, DDX3, DHX9, DHX15, DDX21, DDX24, DHX33, and DHX36) exert broadly antiviral effects against a variety of RNA and DNA viruses [51,53,66–71]. In line with this, we identified the three *Ae. aegypti* DEAD-box helicases Dhx15, AAEL004859, and AAEL008728, to also act as antiviral factors against SINV, DENV, and CHIKV infections. Intriguingly, while these three RNA helicases interact in a protein complex, only *Dhx15* silencing affects the expression of glycolytic genes. These findings suggest that the three DEAD-box helicases are active in various molecular contexts and that regulation of glycolysis by Dhx15 is independent of AAEL004859 and AAEL008728. How the latter two helicases control virus replication in mosquito cells remains to be established. Interestingly, Besson and colleagues recently identified AAEL004859 to strongly bind to the proviral host factor Loquacious in Aag2 cells [34]. Whether AAEL004859 plays an inhibitory function in this context warrants further investigation.

Here, we pursued an in-depth characterization of Dhx15, a highly conserved DEAD-box RNA helicase that was previously characterized as a part of the U2 spliceosome in vertebrates and invertebrates [72]. In human cells, Dhx15 activates MAPK and NFκB signaling during antiviral responses triggered by poly I:C as well as during infection with positive and negative sense RNA viruses [51]. Furthermore, depletion of *Dhx15* increases susceptibility of human cells to infection with a variety of distinct RNA viruses [53], indicating that, similar to our findings, Dhx15 regulates antiviral signaling in the context of a broad range of virus infections. It has been proposed that Dhx15 acts as a RLR co-receptor involved in dsRNA sensing [52,53] making it tempting to speculate that broad antiviral activity both in mammalian and

insect cells is attributed to a conserved role in sensing replicating viral RNA as a common pathogen associated molecular pattern.

Considering its role in controlling an antiviral transcriptional response in mammalian cells, we speculated that *Dhx15* is involved in regulating gene expression in mosquito cells, as well. Indeed, *Dhx15* silencing caused hundreds of genes to be differentially expressed, both in uninfected as well as CHIKV infected cells. However, we did not observe canonical immune genes to be differentially regulated upon *Dhx15* knockdown. Instead, we observed that genes that encode enzymes involved in the core glycolysis pathway were consistently downregulated. This downregulation resulted in reduced lactate levels, suggesting impairment of glycolytic activity in *Dhx15*-depleted mosquito cells. Also in mice, *Dhx15* has previously been linked to energy metabolism but in contrast to our deep-sequencing data, various glycolytic genes were transcriptionally upregulated upon *Dhx15* knockdown in mouse endothelial cells [73]. It is currently unclear what explains this discrepancy but it is likely that overall differences in the metabolic state and integration of other regulatory mechanisms within the two distinct experimental systems account for different metabolic outcomes. It is important to note that it is currently unclear via which signaling cascade Dhx15 regulates glycolytic gene expression in mosquitoes and whether this regulation would be conserved between insects and mammals.

Although the exact mechanism of antiviral activity of Dhx15 remains to be established, we propose that knockdown of *Dhx15*, through the repression of glycolytic genes, establishes a metabolic environment that favors CHIKV infection. Importantly, glycolysis is an important component of cellular energy metabolism as well as a source for biomolecules that act as intermediates for several biosynthesis pathways. For example, the product of the first enzymatic step of glycolysis, glucose-6-phosphate, enters the pentose phosphate pathway, which is responsible for generating pentoses (five-carbon sugars) as well as other RNA and DNA precursors [31]. A shift in glucose availability from energy metabolism to synthesis of biomolecules serves as an attractive model to explain how reduced levels of glycolytic enzymes can control virus growth. We observed a strong downregulation of the rate limiting glycolytic enzyme phosphofructokinase which is expected to make more glucose available for the pentose phosphate pathway. This shift would facilitate an increase in the production of ribonucleotide precursors supporting the higher demand during viral RNA replication. Higher levels of glucose that can feed into the pentose phosphate pathway may also explain the accelerated replication of CHIKV in Aag2 cells cultured in Schneider's medium. In line with this model, Zika virus infection in mosquito cells leads to a shift of glucose fluxes from energy production to five-carbon sugar production, underscoring the relevance of shifting the balance of glucose distribution during virus infections [74]. Importantly, ATP levels are not reduced in *Dhx15* depleted cells (S1A Fig), suggesting that compensatory mechanisms are active to keep cellular energy levels stable. Intriguingly, higher glucose levels in an infectious blood meal enhanced the infection of *Ae. aegypti* mosquitoes with dengue virus, supporting the importance of glucose metabolism for virus infection in adult mosquitoes [75]. In the future, it will therefore be important to investigate the role of Dhx15 in controlling glycolysis and arbovirus replication *in vivo*. Our initial attempts to silence *Dhx15* expression in adult *Ae. aegypti* mosquitoes were thus far unsuccessful in uncovering an antiviral phenotype, possibly because of residual protein activity due to incomplete gene silencing (S6 Fig).

Changes in glycolytic rate are known to occur widely during virus infection, presumably as a consequence of a higher energy and/or nucleotide demand enforced by virus replication [32,33,74,76,77]. On the other hand, an increased energy metabolism has also been shown to activate antiviral defense and glycolytic enhancement is dispensable or even avoided due to triggering of immune responses of the host [31,78,79]. Therefore, the activity of metabolic pathways is likely regulated at multiple levels, potentially explaining why the outcome of

changing metabolic rates appears to be highly specific for distinct virus-host combinations [32,74]. For alphaviruses, increased glycolytic activity has been proposed to support the elevated demand of cellular energy and biomolecules required during Semliki Forest virus, Mayaro virus (MAYV), and SINV replication [76,79–81]. For CHIKV, however, the effect of the virus infection on the metabolic pathways is dependent on the experimental system [32]. CHIKV infection in human cells and in a mouse model incremented cellular metabolism by upregulation of PKM2 and PDHA1, an isoenzyme of pyruvate kinase and a component of the pyruvate dehydrogenase enzyme complex, respectively [82,83]. On the other hand, CHIKV infection lead to downregulation of glycolytic enzymes in a human hepatic cell line [82], similar to what we have observed in *Ae. aegypti* cells.

While in vertebrates, CHIKV virus infections cause a dramatic change in gene expression profiles, largely as a consequence of immune gene induction upon stimulation of interferon signaling [84], we found that CHIKV infection in Aag2 cells only resulted in differential expression of a few dozen genes, the vast majority of which was downregulated. It is currently not clear what causes this curiously weak transcriptional response; two non-mutually exclusive hypotheses are a generally more delicate transcriptional immune signaling in Aag2 cells or active suppression of transcriptional responses by CHIKV and possibly other arboviruses. Despite the modest transcriptional response to CHIKV infection, there was a remarkable overlap of genes that were downregulated upon *Dhx15* silencing and CHIKV infection. It is tempting to speculate that *Dhx15* knockdown creates a metabolic environment that mimics CHIKV infection thereby allowing enhanced virus replication. Altogether, our results uncover an intriguing interaction between transcriptional regulation mediated by a host DEAD-box RNA helicase, alterations in metabolic activities, and antiviral activity in mosquito cells.

## Supporting information

**S1 Fig. RBP candidate genes that control arboviruses replication in mosquito cells. A)** Viability of Aag2 cells was measured using CellTiter-Glo assay after silencing of 15 candidate genes (see Fig 1D and 1E) using the first set of dsRNA. Bars and whiskers represent the mean +/- SD of three independent biological replicates. **B)** Knockdown efficiency of 15 candidate genes (from the experiment shown in Fig 1E) was assessed by RT-qPCR. Bars and whiskers represent the mean +/- SD of three independent biological replicates. Statistical significance was determined using unpaired two tailed t-test (* $p < 0.05$, ** $p < 0.005$, *** $p < 0.0005$). **C)** Levels of SINV were quantified using RT-qPCR after silencing of *Dhx15*, *AAEL008728*, and *AAEL004859* in Aag2 C3PC12 cells. SINV infection was performed with an MOI of 0.1. Bars and whiskers represent the mean +/- SD of three independent biological replicates. Statistical significance was determined using One-Way ANOVA with Holm-Sidak correction (*** $p < 0.0005$). **D)** Knockdown efficiency of genes from panel (*C*) were assessed by RT-qPCR. For each gene, the specific knockdown (light purple) was compared to dsLuc control knockdown (gray). Bars and whiskers represent the mean +/- SD of three independent biological replicates. Statistical significance was determined using unpaired two tailed t-test (* $p < 0.05$). (TIF)

**S2 Fig. The antiviral RNA helicase AAEL004419 is the direct orthologue of Dhx15. A)** Unrooted approximately-maximum likelihood tree of *Drosophila* (purple) and *Ae. aegypti* (brown) RNA-helicases with branch lengths estimated using the CAT approximation described in [50]. **B)** Multiple sequence alignment of *Drosophila* Dhx15 and *Ae. aegypti* AAEL004419. The functional domains (see Fig 2A) are highlighted with colored boxes. **C)** SINV nLuc reporter expression was assessed in Aag2 C3PC12 cells upon transgenic expression of GFP tagged Dhx15. Cells transfected with an empty GFP vector served as negative control.

Bars and whiskers show the mean +/- SD of three biological replicates. Statistical significance was determined using unpaired two tailed t-test (** $p < 0.01$).
(TIF)

**S3 Fig. RNA-sequencing analysis identifies Dhx15 as regulator of glycolysis. A-B)** Levels of CHIKV (**A**) and knockdown efficiency of *Dhx15* (**B**) in samples used for deep-sequencing were assessed by RT-qPCR. CHIKV infection was performed with an MOI of 5. Bars and whiskers represent the mean +/- SD of three independent biological replicates. Statistical significance was determined using unpaired two tailed t-tests (*** $p < 0.0005$). **C)** Number of overlapping genes downregulated (left panel) and upregulated (right panel) upon *Dhx15* silencing in uninfected and CHIKV infected cells. **D)** Schematic representation of the enzymes involved in the glycolysis pathway (left) and $\log_2$ fold change of these genes upon *Dhx15* or firefly luciferase silencing (right) in CHIKV infected cells. **E-F)** Relative lactate concentration upon 2-deoxy-D-glucose (2-DG) treatment in Hela (**E**) and Aag2 (**F**) cells. **G)** CHIKV RNA levels in Aag2 C3PC12 cells infected at an MOI of 0.1 and cultured in L-15 and Schneider's medium, respectively. The viral RNA expression relative to the average levels of housekeeping genes *LAP* and *RpL5* is shown as the mean +/- SD of three independent replicates. **H)** Number of Aag2 C3PC12 cells after sequential *Dhx15* or luciferase control knockdown. Bars and whiskers represent the mean +/- SD of three independent biological replicates. In panels (**E**, **F**, and **H**), statistical significance was determined using unpaired two tailed t-tests (* $p < 0.05$, *** $p < 0.0005$).
(TIF)

**S4 Fig. Differentially regulated genes derived from RNA-sequencing data are specifically dependent on *Dhx15* silencing. A)** Quantification of top five most differentially regulated genes obtained from the RNA-sequencing list of 22 genes with shared downregulation between CHIKV infection and *Dhx15* knockdown (Fig 5B). RNA levels were measured by RT-qPCR after individual silencing of *Dhx15*, *AAEL008728*, and *AAEL004859* in Aag2 C3PC12 cells. Bars and whiskers represent the mean +/- SD of three independent biological replicates. Statistical significance was determined using One-Way ANOVA with Holm-Sidak correction (* $p < 0.05$, *** $p < 0.0005$).
(TIF)

**S5 Fig. CHIKV infection causes reduction of glycolytic gene expression. A)** Relative expression of genes from the glycolysis pathway in CHIKV infected cells (MOI = 5) compared to uninfected cells in control luciferase knockdown conditions. Expression values were extracted from the RNA-sequencing data and normalized to uninfected cells. Bars and whiskers represent the mean +/- SD of three independent biological replicates. Statistics from the DESeq2 analysis are shown (* $P\,adj < 0.05$, ** $P\,adj < 0.005$, *** $P\,adj < 0.0005$).
(TIF)

**S6 Fig. *Dhx15* knockdown is insufficient to uncover antiviral phenotypes *in vivo*. A)** CHIKV replication in individual adult *Ae. aegypti* mosquitoes was assessed after *in vivo* knockdown of *Dhx15* and *Ago2*. Viral RNA levels were quantified by RT-qPCR and normalized to the expression of *LAP*. Relative expression compared to virus levels in dsLuc-injected control mosquitoes is plotted. **B)** Knockdown efficiency was assessed in individual mosquitoes. *Ago2* and *Dhx15* mRNA expression was normalized against *LAP* expression and compared to the mean expression in the dsLuc control condition. Boxplots in (**A**) and (**B**) show median, interquartile range and maximum/minimum values. Outliers are indicated as individual dots and statistical significance was determined using unpaired two tailed t-test (*** p < 0.0005). The

plot was generated in IBM SPSS v25.
(TIF)

**S1 Table. Raw data from targeted knockdown screen and confirmation rounds.** Genes that have been selected for an RNAi screen and, if applicable, updated gene identifiers in the recent version of VectorBase (version 57, accessed April 2022) are shown.
(XLSX)

**S2 Table. Oligonucleotides used in this study.**
(XLSX)

**S3 Table. Differentially expressed genes in RNA-sequencing data.** List 1: *Dhx15* knockdown vs. control knockdown in uninfected cells. List 2: *Dhx15* knockdown vs. control knockdown in CHIKV-infected cells. List 3: CHIKV-infected vs. uninfected cells in control knockdown conditions.
(XLSX)

**S4 Table. Source data file.**
(XLSX)

## Acknowledgments

The authors would like to thank Rebecca Halbach for analyzing expression of DEAD-box helicases in published sequencing data. Thanks also to Ronald van Rij for critical reading of the manuscript. The CHIKV Leiden synthetic LS3 construct was kindly provided by Dr. Martijn J van Hemert at Leiden University Medical Center. Thanks to BEI resources established by the National Institute of Allergy and Infectious Diseases for providing *Aedes aegypti* Liverpool mosquitoes.

## Author Contributions

**Conceptualization:** Samara Rosendo Machado, Pascal Miesen.

**Formal analysis:** Samara Rosendo Machado, Jieqiong Qu, Werner J. H. Koopman, Pascal Miesen.

**Funding acquisition:** Pascal Miesen.

**Investigation:** Samara Rosendo Machado, Jieqiong Qu.

**Methodology:** Samara Rosendo Machado, Jieqiong Qu.

**Project administration:** Pascal Miesen.

**Supervision:** Pascal Miesen.

**Visualization:** Samara Rosendo Machado, Jieqiong Qu.

**Writing – original draft:** Samara Rosendo Machado, Pascal Miesen.

**Writing – review & editing:** Samara Rosendo Machado, Jieqiong Qu, Werner J. H. Koopman, Pascal Miesen.

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
