## [Decision Letter · Decision Letter 0]

20 Jul 2022

Dear Dr. Miesen,

Thank you very much for submitting your manuscript "The DEAD-box RNA helicase Dhx15 controls glycolysis and arbovirus replication in Aedes aegypti mosquito cells" for consideration at PLOS Pathogens. As with all papers reviewed by the journal, your manuscript was reviewed by members of the editorial board and by several independent reviewers. In light of the reviews (below this email), we would like to invite the resubmission of a significantly-revised version that takes into account the reviewers' comments.

We would ask specifically that the authors look into point 2 (major revision) by reviewer 3 on the limitations of the cell titre glo assay, and address this.

We cannot make any decision about publication until we have seen the revised manuscript and your response to the reviewers' comments. Your revised manuscript is also likely to be sent to reviewers for further evaluation.

Sincerely,

Alain Kohl

Associate Editor

PLOS Pathogens

Sara Cherry

Section Editor

PLOS Pathogens

Kasturi Haldar

Editor-in-Chief

PLOS Pathogens

orcid.org/0000-0001-5065-158X

Michael Malim

Editor-in-Chief

PLOS Pathogens

orcid.org/0000-0002-7699-2064

We would ask specifically that the authors look into point 2 (major revision) by reviewer 3 on the limitations of the cell titre glo assay, and address this.

Reviewer's Responses to Questions

**Part I - Summary**

Reviewer #1: Machado and colleagues were able to identify key RBPs that were critical during arbovirus infection using a combination of data mining, RNAi screening, transcriptomics, and experimental assays. From the stringent identification of Ae. aegypti RBPs to the validated RNAi screen, the study highlighted the importance of utilizing genome annotation and proteome repositories in conducting a knockdown screen that is vector-specific. They initially characterized of Ae. aegypti Dhx15 through comparative analysis with orthologs that shed some plausible domain-specific functions of the protein. Establishing the expression and activity of Dhx15 in both mosquito cells and whole organisms also led to preliminary investigations on Dhx15. In addition, multiple co-IP experiments established possible interactions with other identified RBPs. Focusing on the role of Dhx15 during CHIKV infection, they were able to show that regulation of glycolysis by the protein is linked to CHIKV replication. The strength of the study relied on its multiple investigations that clearly supported each other to establish the role of Dhx15. A possible weakness is the study centered CHIKV with only several accounts of using DENV. In addition, the functional experiments involving Dhx15 could have been explored further with other arboviruses. The establishment of Dhx15 over-expressing mosquito cells could also be a possible. Perhaps, these aspects can be investigated in the future.

Reviewer #2: Machado and colleagues have addressed a relevant aspect on antiviral mechanisms in mosquitoes and identified novel antiviral genes in mosquito cells. With the RNA helicase Dhx15 they revealed a potential link between metabolic activity in mosquito cells and antiviral functions. Knockdown of Dhx15 in Aag2 mosquito cells resulted in a reduced expression of glycolytic genes, which was also observed during CHIKV infection. The glyco-metabolic response in mosquito Aag2 cells to CHIKV opens novel research questions for future studies. This is a well-designed study with an in-depth description of the involved methods.

Reviewer #3: The authors underwent a thorough screening of RBP in mosquito cells to identify RBPs that influence multiplication of mosquito borne viruses. Among 461 genes that they screened they identified 15 with antiviral properties. They then pursued a deeper analysis of the function of one of these, Dhx15, and determined that the Dhx15 antiviral effect is driven by glycolysis inhibition, using metabolomic and functional tools. Overall, the study identified Dhx15 as an antiviral factor, determined that this occurs through inhibition of glycolysis, and showed that it is similar to CHIV downregulation of glycolysis.

The study is clearly written and data look solid. However, the overall message suffers from some limitations at the mechanistical level that should be highlighted and discussed. There are two questions that are raised by the study and require additional discussions.

1. It is not clear how Dhx15 regulates glycolysis. The authors suggest that it is through regulation of precursors but they do not provide supportive discussion based on literature for this.

2. It is not clear how glycolysis inhibition reduces CHIKV and other flavivirus infections. My understanding was that glycolysis produces ATP that is required for viral replication. So a reduction in glycolysis should reduce viral replication. The authors should provide further insights here.

While the study clearly identified a new antiviral factors through a screening effort that should be praised, its mechanistic characterization is not complete.

**Part II – Major Issues: Key Experiments Required for Acceptance**

Reviewer #1: These are not mainly major experiments, but some questions/discussions that the authors could address or answer in their revised manuscript. Some suggestions involving additional experiments/analysis may still be performed if the authors wish.

1. Experiments utilized different mosquito cell lines throughout the study. How come? Did persistently infected cells in some assays affected the results? Were you able to determine the transfection efficiency of the dsRNA (single versus double transfections)? In addition, since the RBPs were cloned, knockdown efficiency could also be checked through Western blot.

2. Since you established the key domains of Ae. aegypti Dhx15, perhaps the paper could benefit from a structural model since existing model of the human Dhx15 are available. In your co-IP assays you hypothesized that there could be an RNA-independent interaction between the helicases, were able to test if this is true in the presence of CHIKV viral RNA?

3. You have established that repression of Dhx15 and in turn glycolysis favors CHIKV infection. However in most cases, viral infection demands a high level of energy production and consumption in cells. Does over-expression Dhx15 in mosquito cells favor arbovirus replication? It may also be interesting to identify possible salvage mechanisms that may normalize gylcolysis and allow persistent infection of arboviruses in the absence of this pathway. In your transcriptomics data, were there upregulated genes that interacted with or salvaged the role of Dhx15 during glycolysis? Perhaps you can revisit your data to check for functionally redundant genes that are important and may rescue this mechanism.

Reviewer #2: There are just a few points I would like to suggest:

- Downregulation of phosphofructokinase, aldolase and hexokinase was identified at the transcriptional level after infection with CHIKV/knockdown of Dhx15. Would it be possible to address this at the protein level?

- The switch in medium to Schneider`s medium elevated baseline lactate levels. Downmodulation of glycolytic activity occurred after knockdown of Dhx15. This was discussed to generate a favourable cellular state for CHIKV infection. Thus, infection with CHIKV could be altered in the presence of Schneider`s medium. Do the authors have any indication on virus growth in Aag2 cells during altered substrate supplementation?

- The authors might consider to include the following publication in their discussion: Weng, SC., Tsao, PN. & Shiao, SH. Blood glucose promotes dengue virus infection in the mosquito Aedes aegypti. Parasites Vectors 14, 376 (2021). https://doi.org/10.1186/s13071-021-04877-1. The outlined contribution of glucose to DENV infection supports the relevance of glucose/glycolysis in the context of a virus infection in mosquitoes.

Reviewer #3: Major revisions:

1. It is known that cellular models do not always mimic the in vivo results. Authors should test the effect of Dhx15 in mosquitoes to validate their new antiviral factor.

2. The authors used the CellTiter-Glo kit to quantify cell survival. However this assay is based o ATP measurement, which is influenced by glycolysis, which they claim to modulate with Dxh15 kd. The authors should repeat the cell survival quantification with another assay such as cell counting or house-keeping gene quantification.

**Part III – Minor Issues: Editorial and Data Presentation Modifications**

Reviewer #1: Some inconsistencies in the writing style, formatting, and abbreviations could be addressed.

Reviewer #2: (No Response)

Reviewer #3: Minor revisions:

1. Using a SINV replicon for their screen, the authors missed all effects on virus assembly. This should be mentioned.

2. L. 55. They reference a paper from 1983 to argue that a good understanding of mosquito factors that regulate infection is lacking. They should use a more updated review.

3. L. 106. Please detail how the RBP domains were identified.

4. L. 319. I was surprised to see that Ago2 kd had a moderate effect as compared to other RBPs. Did the authors ensured that the Ago2 kd was sustained through the infection?

5. L. 326. Mention the classes of the other hits, beside from the DED-box RNA helicases

6. L. 433. In the re-analyse of RNAseq, it is not clearly whether the uninfected and CHIKV infected dataset were both transfected with dsRNA control.

7. L. 437. What was the data used for Chi-square? I guess it was between the expression levels of the conditions?

8. L. 447-448. The last sentence of the results section is not supported by data and too speculative. It should be removed.

9. L. 470-474. It is honest for the authors to declare the weaknesses of their study.

10. L. 501. Comparison with human Dhx15 is questionable as Dhx15 deletion did not regulate Toll-compounds or NfkB factors.

11. Fig. 1D and G. Please precise whether the y-axes are log or normal scales. It is confusing when comparing with 1C.

PLOS authors have the option to publish the peer review history of their article (what does this mean?). If published, this will include your full peer review and any attached files.

Reviewer #1: No

Reviewer #2: No

Reviewer #3: No
---

## [Decision Letter · Decision Letter 1]

24 Oct 2022

Dear Dr. Miesen,

Thank you very much for submitting your manuscript "The DEAD-box RNA helicase Dhx15 controls glycolysis and arbovirus replication in Aedes aegypti mosquito cells" for consideration at PLOS Pathogens. As with all papers reviewed by the journal, your manuscript was reviewed by members of the editorial board and by several independent reviewers. The reviewers appreciated the attention to an important topic. Based on the reviews, we are likely to accept this manuscript for publication, providing that you modify the manuscript according to the review recommendations.

Sincerely,

Alain Kohl

Associate Editor

PLOS Pathogens

Sara Cherry

Section Editor

PLOS Pathogens

Kasturi Haldar

Editor-in-Chief

PLOS Pathogens

orcid.org/0000-0001-5065-158X

Michael Malim

Editor-in-Chief

PLOS Pathogens

orcid.org/0000-0002-7699-2064

Reviewer Comments (if any, and for reference):

Reviewer's Responses to Questions

**Part I - Summary**

Reviewer #1: The revised manuscript and rebuttal of the authors have addressed the questions and clarifications raised. Substantial changes to the manuscript and additional supporting experiments and data have been provided.

Reviewer #2: (No Response)

Reviewer #3: The authors made significant efforts to address the comments. Of particular interest is their more detailed hypothetical mechanism of how sugar metabolism alters virus replication by regulating nucleotide production. Also, they tested the effect of Dhx15 knock down in mosquitoes. However, they did not confirm the effect observed in Aag2 cells and attribute it to differences in knock down efficiency. If I am not mistaken, I do not see this new data in the manuscript. The lack of effect of DhX15 in mosquitoes should be included in the manuscript and clearly stated in the text.

**Part II – Major Issues: Key Experiments Required for Acceptance**

Reviewer #1: None.

Reviewer #2: (No Response)

Reviewer #3: The authors should add the new in vivo data in the result section.

**Part III – Minor Issues: Editorial and Data Presentation Modifications**

Reviewer #1: None.

Reviewer #2: (No Response)

Reviewer #3: (No Response)

PLOS authors have the option to publish the peer review history of their article (what does this mean?). If published, this will include your full peer review and any attached files.

Reviewer #1: No

Reviewer #2: No

Reviewer #3: No

Figure Files:

Data Requirements:

Reproducibility:

References:

---

## [Editor Report · Decision Letter 2]

11 Nov 2022

Dear Dr. Miesen,

We are pleased to inform you that your manuscript 'The DEAD-box RNA helicase Dhx15 controls glycolysis and arbovirus replication in Aedes aegypti mosquito cells' has been provisionally accepted for publication in PLOS Pathogens.

Best regards,

Alain Kohl

Academic Editor

PLOS Pathogens

Sara Cherry

Section Editor

PLOS Pathogens

Kasturi Haldar

Editor-in-Chief

PLOS Pathogens

orcid.org/0000-0001-5065-158X

Michael Malim

Editor-in-Chief

PLOS Pathogens

orcid.org/0000-0002-7699-2064
---

## [Editor Report · Acceptance letter]

21 Nov 2022

Dear Dr. Miesen,

We are delighted to inform you that your manuscript, "The DEAD-box RNA helicase Dhx15 controls glycolysis and arbovirus replication in *Aedes aegypti* mosquito cells," has been formally accepted for publication in PLOS Pathogens.

Best regards,

Kasturi Haldar

Editor-in-Chief

PLOS Pathogens

orcid.org/0000-0001-5065-158X

Michael Malim

Editor-in-Chief

PLOS Pathogens

orcid.org/0000-0002-7699-2064